*Correspondence*

**EMBO** *reports*

# Response to Kawamoto et al

Yoshikazu Johmura [1], Teh-Wei Wang[2,3] & Makoto Nakanishi [2✉]

See also: S Kawamoto et al

We have read the study reported by (Kawamoto et al, 2026) which intended to reproduce our previous findings on the senolytic activity of the GLS1 inhibitor BPTES (Johmura et al, 2021) and immune clearance of pre-accumulated senescent cells by anti-PD-1 treatment (Wang et al, 2022) in aged mice. We appreciate the rigor and transparency of their approach and acknowledge the value of such systematic validation across independent groups. Notably, their demonstration that BPTES induces cell death in senescent fibroblasts provides important in vitro support to literatures showing heightened sensitivity of senescent cells to BPTES (Kim et al, 2024; Lee et al, 2024; Takaya et al, 2022; Chen et al, 2025; Li et al, 2025). However, we clarify several interpretive and methodological aspects regarding the senolytic effect of BPTES and the in vivo relevance of their findings.

(1)  The anti-proliferative effects of BPTES on non-senescent cells have already been described in our original report (Johmura et al, 2021). Increased susceptibility to cell death remains the most direct indicator of senolysis (Chang et al, 2016; Zhu et al, 2015). Indeed, in both studies—ours and Kawamoto's—senescent cells exhibited greater sensitivity to BPTES-induced death than their proliferating counterparts (50% vs ~7.5% in IMR-90 cells), indicating that BPTES has senolytic properties in vitro. It is also important to consider that BPTES preferentially targets senescent cells with elevated GLS1 expression. The proportion of such cells may vary depending on the senescence induction method and cell type. In this context, the 7.5–10% cell death reported in non-senescent

controls by Kawamoto et al, may reflect the presence of spontaneously senescent cells, as shown in their EdU incorporation data exhibiting 10–20% non-proliferative cells within these populations. Such pre-existing subpopulations may partially account for observed cell death in the non-senescent control, consistent with a prior report (Lee et al, 2024).

(2)  Regarding the discrepancy in pH modulation, we emphasize that maintaining a stable extracellular pH of 8.5 is technically challenging. We replaced the culture medium every 4 h due to rapid pH alteration caused by $CO_2$ exchange and cellular metabolism. Without continuous monitoring, the extracellular pH could not be stably maintained at 8.5. In addition, as we noted previously, the impact of pH on BPTES-induced cell death is partial and context-dependent, varying with cell types, senescence induction, and culture conditions. Supplementation with α-KG and GSH-MEE also led to a modest reduction in BPTES toxicity in some settings, suggesting that redox imbalance and α-KG depletion may contribute to BPTES sensitivity under certain metabolic states.

(3)  With respect to in vivo experiments, Kawamoto et al, attempted to replicate our findings that both BPTES and anti-PD-1 treatment exert senolytic effects in aged mice. However, we believe that some aspects of their interpretation should be noted. We strongly clarify that any appearance or physical rejuvenation of treated mice was never analyzed or reported in our peer-reviewed publications. No claims regarding visual improvements were made in either of our studies mentioned by Kawamoto et al.

(4)  We advise some points in interpreting grip strength data without accounting for

inter-facility variation. In our facility, we observed a decline in grip strength from ~1.0 N to ~0.6 N between 70–80 weeks of age (Fig. 1; Dataset EV1), which are consistent with our previous results. This decline could depend on the animal facilities tested. Given that baseline values in young mice were not described in Kawamoto's report, it is unclear whether their assessments were suitable to detect functional improvements.

(5)  Analysis of bulk p16INK4a expression in tissues provides only an average signal across a heterogeneous population of cell types. In murine models, unlike in humans, the basal expression level of p16INK4a is extremely low even in aged tissues and often falls below the detection limit in single-cell RNA sequencing (scRNA-seq) analyses. As reported in both Kawamoto's and our studies, this low expression may result in variable qPCR Ct values sensitive to fluctuations arising from sample preparation and tissue handling. Accumulating evidence indicates that p16INK4a expression is not exclusive to senescent cells; rather, it can be upregulated in specific non-senescent cell populations under physiological or pathological contexts (Hall et al, 2017; Helman et al, 2016; Reyes et al, 2022; Safwan-Zaiter et al, 2022; Ogrodnik et al, 2024; Okuma et al, 2017). Therefore, bulk p16INK4a expression may not be a direct surrogate for senescent cell burden or clearance and accurately reflect the dynamics of cellular senescence in vivo. In our previous study, we employed CD26, a cell surface marker for senescent fibroblasts, to isolate and characterize senescent fibroblasts in vivo. The majority of CD26+ fibroblasts in lungs, skins, and adipose tissues exhibited hallmarks of cellular senescence, including lysosomal stress and characteristic senescence-

[1]Division of Cancer and Senescence Biology, Cancer Research Institute, Kanazawa University, Kanazawa, Japan. [2]Division of Cancer Cell Biology, The Institute of Medical Science, The University of Tokyo, Tokyo, Japan. [3]Project Division of Generative AI Utilization Aging Cells, The Institute of Medical Science, The University of Tokyo, Tokyo, Japan.
✉E-mail: mkt-naka@g.ecc.u-tokyo.ac.jp
https://doi.org/10.1038/s44319-026-00752-1 | Published online: 3 April 2026

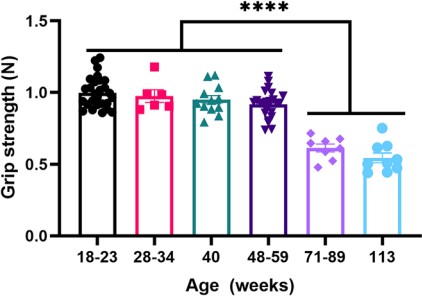

**Figure 1.  Grip strength test results in male mice of different ages.**

Bar plot showing the grip strength of male mice in our animal facility. One-way ANOVA followed by Tukey's test was performed. $N = 28, 6, 12, 24, 8$, and $9$ in each group. All pairwise comparisons between the ≥71-week groups and the ≤59-week groups reached statistical significance with $P$ values $< 0.0001$.

associated transcriptional profiles. In aged mice treated with BPTES, a significant reduction in the proportion of $CD26^+$ fibroblasts was observed in the lungs. Based on these findings, we concluded that BPTES functions as a senolytic agent in vivo.

(6)  In our anti-PD-1 study, we did not analyze bulk $p16^{Ink4a}$ expression levels. Instead, we traced pre-existed $p16^{Ink4a+}$ somatic cells and $PD-L1^+$ subpopulation by fate-mapping approaches (Omori et al, 2020), observing their selective depletion following PD-1 blockade. Since anti-PD-1 therapies activate the immune system that could, paradoxically, induce $p16^{Ink4a}$ expression in non-senescent cells including macrophages, T cells, and others (Hall et al, 2017; Liu et al, 2019; Zhang et al, 2024a), bulk $p16^{Ink4a}$ expression may not reflect the elimination of pre-existed senescent population. As such, the conclusions drawn by Kawamoto et al, regarding the absence of phenotypic or molecular change do not directly contradict our findings but rather highlight the limitations of this analytical approach.

We have recently obtained robust evidence in vivo further supporting the senolytic action of BPTES at single-cell resolution (preprint: Okamura et al, 2025). In this report, we identified a certain fibroblast cluster in the lungs of aged mice, which was selectively eliminated upon BPTES treatment. This cluster displayed various hallmarks of senescence and expressed Dpp4 (CD26), aligning well with our previous findings. We also confirmed that the same cluster was eliminated in DT-treated p16-DTR aged mice (Omori et al, 2020), suggesting that this cluster was p16-positive. In the kidney, BPTES

also selectively eliminates inflammatory proximal tubular cells with the highest GLS1 expression. These results confirmed the senolytic effects of GLS1 inhibitor in vivo, which were supported by many previous literatures (Takaya et al, 2022; Yoshikawa et al, 2022; Lee et al, 2024; Zhang et al, 2024b; Oyama et al, 2024; Kim et al, 2024; Chen et al, 2025). Consistent with this, the therapeutic potential of GLS1 inhibition has also been widely reported on age-related disorders (Harvey et al, 2025; Kono et al, 2019; Simon et al, 2020; Du et al, 2018; Cui et al, 2019; Bertero et al, 2016; Li et al, 2025; Chen et al, 2026).

Taken together, these findings underscore important methodological considerations: assessing bulk $p16^{INK4a}$ expression at the tissue level may not quantify an exact senescent cell burden. Unbiased scRNA-seq offers a powerful alternative, enabling the identification of specific senescent populations that are selectively eliminated by a given compound in vivo. Such an approach not only confirms the senolytic efficacy of candidate drugs at single-cell resolution but also reveals the molecular and functional characteristics of the targeted senescent cell subsets. These insights are critical for elucidating the mechanisms underlying senescence-dependent inflammaging in vivo.

## Methods

### Grip strength test

All C57BL/6 male mice were housed 2–5 to a cage at an ambient temperature of 23–25 °C in a humidity-controlled room and were maintained on a 12-h light/dark cycle (08:00 to 20:00 light on) with standard food (CA-1, CLEA) and water provided ad libitum. All animals were handled according to the

Guidelines for Animal Experiments of the Institute of Medical Science, the University of Tokyo, and Institutional Laboratory Animal Care. All animal experiments were approved by the Animal Experiment Committee at IMSUT (A16-33, A21-26). Grip strength was measured using a BIO-GS3 grip strength testing device (Bioseb). Mice were held by the tail and allowed to grasp the metal grid with their forepaws. The examiner then gently pulled the mouse backward in the horizontal plane until it released the grid, at which point the maximum force displayed on the device was recorded. Each mouse was tested three times, and the average of the three trials was used as the final value for statistical analysis.

## Peer review information

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

## Acknowledgements

This study was supported by the Japan Agency for Medical Research and Development (AMED) under grant no. JP23zf0127003h (M.N.).

## Author contributions

**Yoshikazu Johmura**: Writing—original draft; Writing—review and editing. **Teh-Wei Wang**: Writing—original draft; Writing—review and editing. **Makoto Nakanishi**: Writing—original draft; Writing—review and editing.

## Disclosure and competing interests statement

MN is a scientific advisor and a shareholder of reverSASP Therapeutics.

