## [Peer Review File · EMBO Reports]

Response to Kawamoto et al

Yoshikazu Johmura, Teh-Wei Wang, and Makoto Nakanishi

Corresponding author(s): Makoto Nakanishi (mkt-naka@g.ecc.u-tokyo.ac.jp)

Review Timeline:

Submission Date:	2nd Jul 25
Editorial Decision:	29th Aug 25
Revision Received:	16th Sep 25
Accepted:	10th Mar 26

Transaction Report:

Dear Makoto,

Thank you for the submission of your Correspondence article, responding to Kawamoto et al, which is under consideration at EMBO reports. We have now received the three enclosed reports on it.

As you will see, all three referees consider your argumentation insightful and valid. We will therefore proceed with its publication, after a few editorial points have been addressed:

- All corresponding authors must provide an ORCID ID. This information is currently missing.
- Please add a 'Disclosure and competing interests statement'. For more information see <https://www.embopress.org/page/journal/14693178/authorguide#conflictsofinterest>
- You can add an Acknowledgment section, if needed. Information on funding, if this applies, can be listed in the Acknowledgments and need to be entered in the online manuscript tracking system. Please ensure that these match each other.
- Please provide Figure 1 as a separate high quality figure file. As the figure reports on actual data, we would please need the source data used to generate the graph as .xls file and a short Methods section describing how grip strength was measured, the genotype, housing and husbandry conditions of the mice and an ethics approval. (Please have a look at this recent Correspondence, which also contained data and a methods section, to give you an idea what it looks like:
<https://www.embopress.org/doi/epdf/10.1038/s44318-025-00479-0>
- Please reformat the references: et al needs to be used after 10 author names; DOIs should only be used for preprints and datasets that have not been published yet.
- As title I suggest "Response to Kawamoto et al". Please also add "Reply to: Kawamoto et al" to the title page.
- The related manuscript by Okamura et al was available for the referees. As discussed, please ensure its bioRxiv deposition before publication of the Correspondence, so that it is available to all readers. Please note the reference format for preprints: in the text citations need the prefix preprint (preprint: Okamura et al, 2025). In the reference list it is followed by the tag [PREPRINT].

I am looking forward to receiving a revised manuscript. Please do not hesitate to contact me if you have any comments or questions.

Kind regards,

Martina

=====

Referee #1:

This is a timely and important topic, given the growing interest in senolytic interventions and their potential clinical applications, as well as the broader issue of reproducibility in scientific research. The authors raise valid and insightful points regarding differences between studies and provide thoughtful explanations for the variability in results. I appreciate the constructive and respectful tone throughout the piece.

As a correspondence article reflecting the authors perspectives, I believe it should be published without revisions, pending, of course, the publication of the manuscript from the Hara's lab

Referee #2:

I have carefully read Dr. Nakanishi and colleagues' thoughtful and systematic response to the concerns raised by Ejihara et al. regarding the senolytic activity of GLS1 inhibition and anti-PD-1 therapy in aged mice. Their point-by-point clarification is rigorous and provides valuable methodological and interpretive insights.

I particularly appreciate the emphasis on several important aspects:

1. Context-dependent senolysis - Both positive and negative outcomes with BPTES highlight that senescent cell susceptibility is highly dependent on the cell type, senescence inducer, and tissue context. This debate itself illustrates the complexity and heterogeneity of senescent cells, and it is unlikely that any single "pan-senolytic" will target all senescent populations.
2. Limitations of bulk p16INK4a measurements - I agree with Dr. Nakanishi's point that bulk p16INK4a expression alone is insufficient to definitively conclude senescent cell clearance or suppression. Integrating multiple senescence markers and single-cell-level analyses will provide a more accurate and biologically meaningful assessment.
3. Value of both positive and negative findings - The combination of rigorous replication, context-specific results, and mechanistic clarification is critical for moving the field forward. Studies reporting either positive or negative outcomes both contribute to refining senolytic strategies and understanding the heterogeneity of senescent cell biology.

Going forward, I strongly support the suggestion that future reports on new senolytics should clearly define the experimental context-including the senescence induction method, cell type, and tissue environment-in which selective senolysis is observed. This approach will facilitate reproducibility, meaningful interpretation, and ultimately the rational translation of senolytic interventions.

Referee #3:

The commentary addresses convincingly most issues raised by comparison of the Johmura and Kawamoto results. Further potential issues have been indicated as points 1-3 of my review of the Kawamoto paper.

The authors have addressed all minor editorial requests.

Prof. Makoto Nakanishi
University of Tokyo
Institute of Medical Science
4-6-1 Shirokanedai, Minatoku
Tokyo, Tokyo 108-8639
Japan

Dear Makoto,

Thank you for the submission of your revised Correspondence article. I have now uploaded the further revised and final manuscript text and accepted it for publication.

Your article will be published back-to-back with the article from Kawamoto et al.

Your manuscript will be copy edited and you will receive page proofs prior to publication. Please note that you will be contacted by Springer Nature Author Services to complete licensing information.

Please contact me any time of you have any questions.

Kind regards,

Martina
